# Approximating Nash Equilibria in General-Sum Games via Meta-Learning

## Abstract

Nash equilibrium is perhaps the best-known solution concept in game theory. Such a solution assigns a strategy to each player which offers no incentive to unilaterally deviate. While a Nash equilibrium is guaranteed to always exist, the problem of finding one in general-sum games is PPAD-complete, generally considered intractable. Regret minimization is an efficient framework for approximating Nash equilibria in two-player zero-sum games. However, in general-sum games, such algorithms are only guaranteed to converge to a coarse-correlated equilibrium (CCE), a solution concept where player can correlate their strategies. In this work, we use meta-learning to minimize the correlations in strategies produced by a regret minimizer. This encourages the regret minimizer to find strategies that are closer to a Nash equilibrium. The meta-learned regret minimizer is still guaranteed to converge to a CCE, but we give a bound on the distance to Nash equilibrium in terms of our meta-loss. We evaluate our approach in general-sum imperfect information games. Our algorithms provide significantly better approximations of Nash equilibria than state-of-the-art regret minimization techniques.

## 1. Introduction

The Nash equilibrium is one of the most influential solution concepts in game theory. A strategy profile is a Nash equilibrium if it has the guarantee that no player can benefit by unilaterally deviating from it. The robustness of this guarantee means that Nash equilibria have applications in many domains ranging from economics (Vickrey, 1961; Milgrom & Weber, 1982) to machine learning (Goodfellow et al., 2014). Finding an efficient algorithm for computing Nash equilibria has attracted much attention (Rosenthal,

1973; Monderer & Shapley, 1996; Kearns et al., 2001; Cai & Daskalakis, 2011; Littman & Stone, 2005). However, it was shown that, in its full generality, finding a Nash equilibrium is PPAD-complete (Papadimitriou, 1994; Daskalakis et al., 2009a). Many related decision problems, such as 'Is a given action in the support of a Nash equilibrium?', are NP-complete (Gilboa & Zemel, 1989).

Despite these negative results, computing Nash equilibria in special classes of games, in particular two-player zero-sum games, is tractable. In this setting, regret minimization has become the dominant approach for finding Nash equilibria (Nisan et al., 2007). This framework casts each player as an independent online learner who repeatedly interacts with the game, selecting strategies according to dynamics that lead to sublinear growth of their accumulated *regret*. Regret minimizers guarantee convergence to Nash equilibria in two-player zero-sum games, and are the basis for many significant results in imperfect information games (Bowling et al., 2015; Moravčik et al., 2017; Brown & Sandholm, 2018; Brown et al., 2020; Brown & Sandholm, 2019a; Schmid et al., 2023).

Outside the two-player zero-sum setting, regret minimization algorithms are no longer guaranteed to converge to a Nash equilibrium. Instead, a regret minimizer's empirical distribution of play converges to a coarse-correlated equilibrium (CCE) (Hannan, 1957; Hart & Mas-Colell, 2000). The CCE is a relaxed equilibrium concept, which gives a distribution over the *outcomes* of the game such that it isn't beneficial for any player to deviate from it. If this distribution is uncorrelated, meaning it can be expressed as a profile of independent strategies, it is also a Nash equilibrium. As such, Nash equilibria form a subset of CCEs, for which the outcome distribution can be marginalized into strategies of the individual players. The degree to which a CCE is correlated, or how much a player can infer about the actions of other players given their action, can be formalized by total correlation (Watanabe, 1960).

A recently proposed *learning not to regret* framework allows one to meta-learn a regret minimizer to optimize a specified objective, while keeping regret minimization guarantees (Sychrovský et al., 2024). Their goal was to accelerate the empirical convergence rate on a distribution of black-box tasks. In this work, we meta-learn predictions

[1]Anonymous Institution, Anonymous City, Anonymous Region, Anonymous Country. Correspondence to: Anonymous Author <anon.email@domain.com>.

Preliminary work. Under review by the International Conference on Machine Learning (ICML). Do not distribute.

that optimize an alternative meta-objective: minimizing correlation in the players' strategies. The resulting algorithm is still guaranteed to converge to a CCE, and is meta-learned to empirically converge to a Nash equilibrium on a distribution of interest. If the support of the distribution doesn't include all general-sum games, the problem of finding Nash may be tractable even if P≠PPAD. We further show this approach is sound by providing a bound on the distance to a Nash equilibrium in terms of our meta-objective. We evaluate our approach in general-sum imperfect information games. Our algorithms provide significantly better approximations of Nash equilibria than state-of-the-art regret minimization techniques.

## 1.1. Related Work

The Nash equilibrium is one of the oldest solution concepts in game theory. Thanks to its many appealing properties, developing efficient algorithms for approximating Nash equilibria has seen much attention (Kontogiannis et al., 2009; Daskalakis et al., 2009b; 2007; Bosse et al., 2010; Deligkas et al., 2023; Li et al., 2024). Furthermore, it was shown that, unless P = NP, polynomial algorithms for finding all Nash equilibria cannot exist (Gilboa & Zemel, 1989). This negative result suggests that there are games for which finding a Nash equilibrium requires enumerating all possible strategies — an amount exponential in the number of actions.

The Lemke-Howson algorithm (Lemke & Howson, 1964) is one such algorithm, which provably finds a Nash equilibrium of two-player general-sum games in normal-form. It works by constructing a path on an abstract polyhedron, which is guaranteed to terminate at the Nash equilibrium. Similar to the simplex method (Murty, 1984), the path may be exponentially long in some games. However, such games are empirically rare (Codenotti et al., 2008). Several modifications of the Lemke-Howson algorithm were proposed to improve its empirical performance (Codenotti et al., 2008; Gatti et al., 2012). However, the algorithm cannot work with games in extensive-form. When converted to normal-form, the size of the game increases exponentially, making these algorithms scale very poorly.

Regret minimization is a powerful framework for online convex optimization (Zinkevich, 2003; Nisan et al., 2007), with regret matching as one of the most popular algorithms in game applications (Hart & Mas-Colell, 2000). Counterfactual regret minimization enables the use of regret matching in sequential decision-making, by decomposing the full regret to individual states (Zinkevich et al., 2007). In two-player zero-sum games, regret minimization algorithms are guaranteed to converge to a Nash equilibrium. Many prior works explored modifications of regret matching to speed up its empirical performance in two-player zero-sum games, such as CFR$^+$ (Tammelin, 2014), Linear

CFR (Brown et al., 2019), PCFR$^+$ (Farina et al., 2023), Discounted CFR (Brown & Sandholm, 2019c), and their hyperparameter-scheduled counterparts (Zhang et al., 2024).

Despite the lack of theoretical guarantees in general-sum games, regret minimization algorithms empirically converge close to Nash equilibria on many standard benchmarks (Risk & Szafron, 2010; Gibson, 2014; Brown & Sandholm, 2019a). Recently, some theoretical advancements have been made to understand this empirical performance. If the game has a special 'pair-wise zero-sum' structure, then the regret minimizers are guaranteed to find a Nash equilibrium (Cai & Daskalakis, 2011). Moreover, if a game is 'close' to such 'pair-wise zero-sum' games, the regret minimzers converge 'close' to a Nash equilibrium (MacQueen & Wright, 2024).

A recently introduced extension of regret matching, predictive regret matching (Farina et al., 2021), forms a continuous class of algorithms with regret minimization guarantees. Subsequently, (Sychrovský et al., 2024) introduced the 'learning not to regret' framework—a way to meta-learn the predictions while keeping regret minimization guarantees. Their aim was to accelerate convergence on a class of oblivious environments.

## 1.2. Main Contribution

In this work, we extend the *learning not to regret* framework to encourage convergence to Nash equilibria in general-sum games. Our approach penalizes correlations in the average empirical strategy profile found by the regret minimizer. While our meta-learned algorithms do not guarantee convergence to a Nash equilibrium, we find that our algorithms empirically converge to CCEs with low correlations in the players' strategies, and provide significantly better approximations of Nash equilibria than prior regret minimization algorithms.

We demonstrate the feasibility of our approach by conducting experiments in multiplayer general-sum games. We start with a distribution of normal-form games, where prior regret minimization algorithms overwhelmingly converge to a strictly correlated CCE. Next, we shift our attention to Leduc poker, a standard extensive-form imperfect information benchmark. We show that, after a small modification of the rules (to make the game general-sum), prior regret minimizers no longer reliably converge to a Nash equilibrium. When trained on this distribution, our meta-learning framework produces a regret minimizer that reach significantly closer to a Nash equilibrium. Finally, we demonstrate that our framework can even be used to obtain better approximations of a Nash equilibrium on a single general-sum game rather than just a family of games. We choose the three-player Leduc poker, obtaining, to our best knowledge, the closest approximation of a Nash equilibrium of this game.

## 2. Preliminaries

We briefly introduce the formalism of incomplete information games we will use. Next, we describe regret minimization, a general online convex optimization framework. Finally, we discuss how regret minimization can be used to find equilibria of these games.

### 2.1. Games

We work within a formalism based on factored-observation stochastic games (Kovařík et al., 2022) with terminal utilities.

**Definition 2.1.** A game is a tuple $\langle \mathcal{N}, \mathcal{W}, w^o, \mathcal{A}, \mathcal{T}, u, \mathcal{O} \rangle$, where

- $\mathcal{N} = \{1, \ldots, n\}$ is a **player set**. We use symbol $i$ for a player and $-i$ for its opponents.
- $\mathcal{W}$ is a set of **world states** and $w^0 \in \mathcal{W}$ is a unique initial world state.
- $\mathcal{A} = \mathcal{A}_1 \times \cdots \times \mathcal{A}_n$ is a space of **joint actions**. A world state with no legal actions is **terminal**. We denote the set of terminal world states as $\mathcal{Z}$.
- After taking a (legal) joint action $a$ at $w$, the **transition function** $\mathcal{T}$ determines the next world state $w'$, drawn from the probability distribution $\mathcal{T}(w, a) \in \Delta(\mathcal{W})$.
- $u_i(z)$ is the **utility** player $i$ receives when a terminal state $z \in \mathcal{Z}$ is reached.
- $\mathcal{O} = (\mathcal{O}_1, \ldots, \mathcal{O}_n)$ is the **observation function** specifying both the private and public observation that players receives upon the state transition.

The space $\mathcal{S}_i$ of all action-observation sequences can be viewed as the infostate tree of player $i$. A **strategy profile** is a tuple $\boldsymbol{\sigma} = (\boldsymbol{\sigma}_1, \ldots, \boldsymbol{\sigma}_n)$, where each player's **strategy** $\boldsymbol{\sigma}_i : s_i \in \mathcal{S}_i \mapsto \boldsymbol{\sigma}_i(s_i) \in \Delta^{|\mathcal{A}_i(s_i)|}$ specifies the probability distribution from which player $i$ draws their next action conditional on having information $s_i$. We denote the space of all strategy profiles as $\boldsymbol{\Sigma}$. A **pure strategy** $\boldsymbol{\rho}_i$ is a deterministic strategy: i.e. $\sigma_i(s_i, a_i) = 1$ for some $a_i \in \mathcal{A}_i(s_i)$. A selection of pure strategies for all players $\boldsymbol{\rho} = (\boldsymbol{\rho}_1, \ldots \boldsymbol{\rho}_n)$ is a **pure strategy profile** and the set of all pure strategy profiles is $\mathbf{P}$.

Let $\Delta(X)$ denote the set of distributions over a domain $X$. A **joint strategy profile** $\boldsymbol{\delta} \in \Delta(\mathbf{P})$ is a distribution over pure strategy profiles. As such, every strategy profile is also a joint strategy profile. However, the opposite is not true in general: only *some* joint strategy profiles are "marginalizable" into an equivalent strategy profile, while those with correlations between players' strategies are not.

The expected **utility** under a joint strategy profile $\boldsymbol{\delta}$ is $u_i(\boldsymbol{\delta}) = \mathbb{E}_{z \sim \boldsymbol{\delta}} \, u_i(z)$, where the expectation is over the

terminal states $z \in \mathcal{Z}$ and their reach probability under $\boldsymbol{\delta}$. The **best-response** to the joint strategy of the other players is $br(\boldsymbol{\delta}_{-i}) \in \arg\max_{\boldsymbol{\sigma}_i} u_i(\boldsymbol{\sigma}_i, \boldsymbol{\delta}_{-i})$, where $\boldsymbol{\delta}_{-i}(\boldsymbol{\rho}_{-i}) = \sum_{\boldsymbol{\rho}_i \in \mathcal{A}_i} \boldsymbol{\delta}(\boldsymbol{\rho}_i, \boldsymbol{\rho}_{-i})$.

We may measure the distance of a strategy profile $\boldsymbol{\sigma}$ from a Nash equilibrium by its **NashGap**: the maximum gain any player can obtain by unilaterally deviating from $\boldsymbol{\sigma}$

$$\text{NashGap}(\boldsymbol{\sigma}) = \max_{i \in \mathcal{N}} \left[ u_i(br(\boldsymbol{\sigma}_{-i}), \boldsymbol{\sigma}_{-i}) - u_i(\boldsymbol{\sigma}) \right].$$

A strategy profile is a Nash equilibrium if its NashGap is zero.[1]

The coarse correlated equilibrium (CCE) (Moulin & Vial, 1975; Nisan et al., 2007) is a generalization of Nash equilibrium to joint strategy profiles that allows for correlation between players' strategies. A CCE is a joint strategy profile such that any unilateral deviation by any player doesn't increase that player's utility, while other players continue to play according to the joint strategy. We define the **CCE Gap** as

$$\text{CCE Gap}(\boldsymbol{\delta}) = \max_{i \in \mathcal{N}} \left[ u_i(br(\boldsymbol{\delta}_{-i}), \boldsymbol{\delta}_{-i}) - u_i(\boldsymbol{\delta}) \right].$$

A joint strategy profile $\boldsymbol{\delta}$ is a CCE if and only if its CCE Gap is zero. If a joint strategy profile has zero CCE Gap, and can be written in terms of its marginal strategies for each player $\boldsymbol{\delta} = (\boldsymbol{\sigma}_1, \ldots, \boldsymbol{\sigma}_n)$, then its marginals $\boldsymbol{\sigma}_i$ are a Nash equilibrium. In general, CCEs do not admit this player-wise decomposition of the joint strategy profile—see Section 4.1 for an example.

### 2.2. Regret Minimization

An **online algorithm** $m$ for the regret minimization task repeatedly interacts with an **environment** through available actions $\mathcal{A}_i$. The goal of a regret minimization algorithm is to maximize its hindsight performance (i.e., to minimize regret). For reasons discussed in the following section, we will describe the formalism from the point of view of player $i$ acting at an infostate $s \in \mathcal{S}_i$.

Formally, at each step $t \leq T$, the algorithm submits a **strategy** $\boldsymbol{\sigma}_i^t(s) \in \Delta^{|\mathcal{A}_i(s)|}$. Subsequently, it observes the expected **reward** $\boldsymbol{x}_i^t \in \mathbb{R}^{|\mathcal{A}_i(s)|}$ at the state $s$ for each of the actions from the environment, which depends on the strategy in the rest of the game. The difference in reward obtained under $\boldsymbol{\sigma}_i^t(s)$ and any fixed action strategy is called the **instantaneous regret** $\boldsymbol{r}_i(\boldsymbol{\sigma}^t, s) = \boldsymbol{x}_i^t(\boldsymbol{\sigma}^t) - \langle \boldsymbol{\sigma}_i^t(s), \boldsymbol{x}_i^t(\boldsymbol{\sigma}^t) \rangle \mathbf{1}$. The **cumulative regret** throughout time $t$ is $\boldsymbol{R}_i^t(s) = \sum_{\tau=1}^{t} \boldsymbol{r}_i(\boldsymbol{\sigma}^\tau, s)$.

The goal of a regret minimization algorithm is to ensure that the regret grows sublinearly for any sequence of re-

---

[1]This is because then the individual strategy profiles are mutual best-responses.

wards. One way to do that is for $m$ to select $\boldsymbol{\sigma}_i^{t+1}(s)$ proportionally to the positive parts of $\boldsymbol{R}_i^t(s)$, known as regret matching (Blackwell et al., 1956).

### 2.3. Connection Between Games and Regret Minimization

In normal-form games, or when $\mathcal{S}_i$ is a singleton, if the **external regret** $R_i^{\text{ext},T} = \max_{a \in \mathcal{A}_i} \boldsymbol{R}_i^T$ grows as $\mathcal{O}(\sqrt{T})$ for all players, then the empirical average joint strategy profile $\overline{\boldsymbol{\delta}}^T \stackrel{\text{def}}{=} \frac{1}{T} \sum_{t=1}^T \boldsymbol{\sigma}_1^t \times \cdots \times \boldsymbol{\sigma}_n^t$ converges to a CCE as $\mathcal{O}(1/\sqrt{T})$ (Nisan et al., 2007).

In extensive-form games, in order to obtain the external regret, we would need to convert the game to normal-form. However, the size of the normal-form representation is exponential in the size extensive-form representation. Thankfully, one can upper-bound the normal-form regret by individual (i.e. per-infostate) **counterfactual regrets** (Zinkevich et al., 2007)

$$\sum_{i \in \mathcal{N}} R_i^{\text{ext},T} \leq \sum_{i \in \mathcal{N}} \sum_{s \in \mathcal{S}_i} \max \left\{ \left\| \boldsymbol{R}_i^T(s) \right\|_\infty, 0 \right\}.$$

The counterfactual regret is defined with respect to the **counterfactual reward**. At an infostate $s \in \mathcal{S}_i$, the counterfactual rewards measure the expected utility the player would obtain in the game when playing to reach $s$. In other words, it is the expected utility of $i$ at $s$, multiplied by the opponent's and chance's contribution to the probability of reaching $s$. We can treat each infostate as a separate environment, and minimize their counterfactual regrets independently. This approach converges to a CCE (Zinkevich et al., 2007).

In two-player zero-sum games, the empirical average strategy $\overline{\boldsymbol{\sigma}}$ is guaranteed to converge to a Nash equilibrium (Zinkevich et al., 2007). In fact, any CCE of a two-player zero-sum game is guaranteed to be marginalizable (Nisan et al., 2007). Intuitively, any correlations will be beneficial for one of the players, which makes it irrational for the opponent to follow it.

## 3. Meta-Learning Framework

We aim to find a regret minimization algorithm $m_\theta$ with some parameterization $\theta$ which tends to converge close to a Nash equilibrium on a distribution of games $G$. In this section, we describe the predictive regret minimization algorithm over which we meta-learn. Then, we formalize our optimization objective for the meta-learning.

### 3.1. Neural Predictive Counterfactual Regret Minimization (NPCFR)

We work in the learning not to regret framework (Sychrovský, 2024), which is built on the predictive regret

---

**Algorithm 1** Neural Predictive Regret Matching (Sychrovský et al., 2024)

1: $\boldsymbol{R}^0 \leftarrow \boldsymbol{0} \in \mathbb{R}^{|A|}, \quad \boldsymbol{x}^0 \leftarrow \boldsymbol{0} \in \mathbb{R}^{|A|}$
2: $\boldsymbol{e}_s \leftarrow$ embedding of state $s$

3: NextStrategy()
4:      $\boldsymbol{\xi}^t \leftarrow [\boldsymbol{R}^{t-1} + \boldsymbol{p}^t]^+$
5:      **if** $\|\boldsymbol{\xi}^t\|_1 > 0$
6:          $\boldsymbol{\sigma}^t \leftarrow \boldsymbol{\xi}^t / \|\boldsymbol{\xi}^t\|_1$
7:      **else**
8:          $\boldsymbol{\sigma}^t \leftarrow$ arbitrary point in $\Delta^{|A|}$
9:      **return** $\boldsymbol{\sigma}^t$

10: ObserveReward($\boldsymbol{x}^t, \boldsymbol{e}_s$)
11:      $\boldsymbol{r}^t \leftarrow \boldsymbol{r}(\boldsymbol{\sigma}^t, \boldsymbol{x}^t)$
12:      $\boldsymbol{R}^t \leftarrow \boldsymbol{R}^{t-1} + \boldsymbol{r}^t$
13:      $\boldsymbol{p}^{t+1} \leftarrow \alpha(\boldsymbol{r}^t + \pi(\boldsymbol{r}^t, \boldsymbol{R}^t, \boldsymbol{e}_s | \theta))$

---

matching (PRM) (Farina et al., 2021). PRM is an extension of regret matching (Hart & Mas-Colell, 2000) which additionally uses a predictor about future reward. PRM provably enjoys $\mathcal{O}(\sqrt{T})$ bound on the external regret for arbitrary bounded predictions (Farina et al., 2021).

Neural predictive regret matching is an extension of PRM which uses a predictor $\pi$, parameterized by a neural network $\theta$ (Sychrovský et al., 2024); see Algorithm 1. At each step $t$ and each infostate $s \in \mathcal{S}_i, i \in \mathcal{N}$, the predictor $\pi(\cdot | \theta)$ makes a prediction about the next observed regret $\boldsymbol{r}^{t+1}$. This prediction is then used when selecting the strategy, as if that regret was in fact observed. The strategy is then selected as if this predicted regret was observed. Network parameters $\theta$ are shared across all infostates $s \in \mathcal{S}_i, i \in \mathcal{N}$, and $\alpha \in \mathbb{R}$ is a hyperparameter, see Appendix B for more details. The $\boldsymbol{e}_s$ denotes some embedding of the infostate $s$; see Section 4.

Since we make the predictions bounded, the predictor can be meta-learned to minimize a desired objective while maintaining the regret minimization guarantees (Sychrovský et al., 2024), which makes the algorithm converge to a CCE. We use a novel meta-objective, which is introduced in the following section, to encourage the algorithm to converge to a Nash equilibrium. Applying the algorithm to counterfactual regrets at each infostate allows us to use it on extensive-form games. This setup is refer to as neural predictive counterfactual regret minimization (NPCFR).

### 3.2. Meta-Loss Function

Any instance of NPCFR is a regret minimizer and is therefore guaranteed to converge to a CCE. Since any Nash equilibrium is a CCE for which player strategies are uncorrelated, we propose a meta-loss objective that penalizes correlation

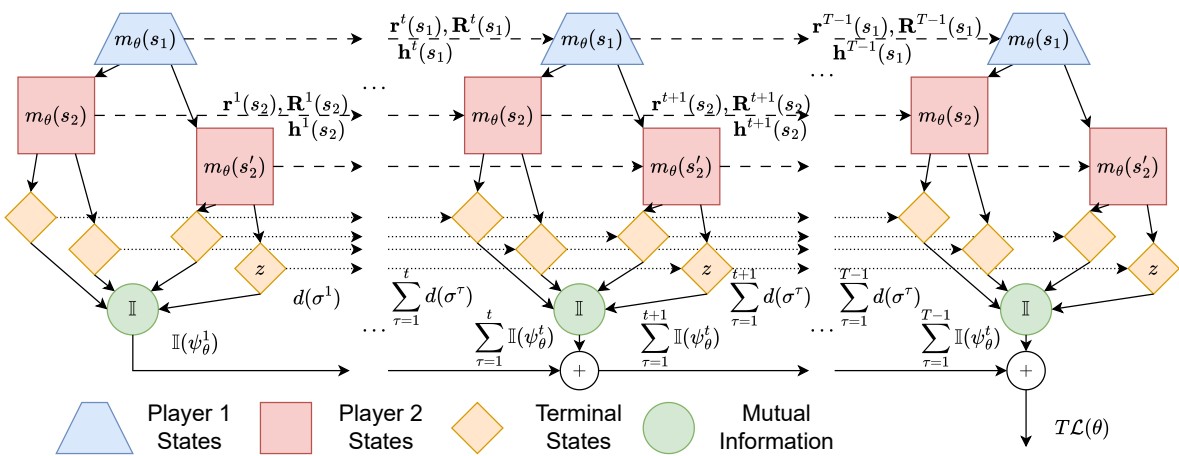

*Figure 1.* Computational graph of NPCFR$^{(+)}$ for a simple extensive form game. The algorithm $m_\theta$ produces a strategy in each infostate using the regret $\mathbf{r}^t$, $\mathbf{R}^t$, and its hidden state $\mathbf{h}^t$, see Algorithm 1. Each terminal state $z \in \mathcal{Z}$ accumulates its empirical average reach probability $\frac{1}{t} \sum_{\tau=1}^{t} d(\boldsymbol{\sigma}^\tau)(z)$. Marginalizability $\mathbb{I}$ is computed between this accumulated average reach and the reach probability under the empirical average strategy profile in the game tree. The meta-loss is the average mutual information experienced over $T$ steps, according to (1). Its gradient is propagated through all edges.

in the CCE found by NPCFR. Informally, these correlations measure the mutual dependence of players' strategies. Or in other words, how much a player can infer about the actions of other players given their action.

One could express this measure of correlation as the *mutual information* of the CCE.[2] However, for extensive-form games, this leads to an exponential blow-up in the size of the game, since there are exponentially more pure strategies than infostates. Instead, we exploit the structure of extensive-form games to define an equivalent meta-loss that does not suffer from this blow-up.

Formally, let $\psi^T = (\boldsymbol{\sigma}^t)_{t=1}^T$ be a sequence of strategy profiles selected by a regret minimizer. Let $d(\boldsymbol{\sigma})$ be the distribution of reach probabilities of terminals $z \in \mathcal{Z}$ under $\boldsymbol{\sigma}$, where $d(\boldsymbol{\sigma})(z)$ is the reach probability of $z$. $d(\boldsymbol{\sigma})$ can be decomposed into a product of player's (and chance's) contribution of reaching $z$: $d(\boldsymbol{\sigma})(z) = d_c(z) \prod_{i \in \mathcal{N}} d_i(\boldsymbol{\sigma})(z)$ where $d_c(z)$ is chance's contribution to reaching $z$ and $d_i(\boldsymbol{\sigma})(z)$ is the product of $\sigma_i(s,a)$ for infostates $s \in \mathcal{S}_i$ on the path to $z$.

The average distribution over terminals across $\psi^T$ is $d(\psi^T) \stackrel{\text{def}}{=} \frac{1}{T} \sum_{t=1}^T d(\boldsymbol{\sigma}^t)$. We define the *marginal across terminals* $\mu(\psi^T)$ for $\psi^T$ as a distribution across terminals under the empirical average strategy in the game. Formally,

$$\mu(\psi^T)(z) \stackrel{\text{def}}{=} d_c(z) \prod_{i \in \mathcal{N}} \frac{1}{T} \sum_{t=1}^T d_i(\boldsymbol{\sigma}^t)(z).$$

---

[2]We describe this measure in more detail in Appendix A.

In words, this is the distribution on terminals induced by each player's empirical average strategy in the game tree. The sequence $\psi^T$ is uncorrelated if $d(\psi^T)$ and $\mu(\psi^T)$ have no mutual dependence. This is formally captured by taking the KL divergence across terminals between $d(\psi^T)$ and $\mu(\psi^T)$. We denote this KL as $\mathbb{I}(\psi)$, since it is equal to mutual information for the two-player case and total correlation for the $n$-player case (Watanabe, 1960).

**Definition 3.1.** We say that $\psi^T$ is $\epsilon$-extensive-form marginalizable ($\epsilon$-EFM) if

$$\mathbb{I}(\psi^T) \stackrel{\text{def}}{=} D_{\text{KL}}\left(d(\psi^T) \,||\, \mu(\psi^T)\right) \leq \epsilon. \tag{1}$$

When a sequence of strategies of a regret minimizer is close to extensive-form marginalizable, it provably converges close to a Nash equilibrium. Formally, let $\overline{\boldsymbol{\sigma}}^T$ be the average strategy profile of $\psi^T$.

**Theorem 1.** *If $\psi^T$ was produced by an external regret minimizer with regret bounded by $\mathcal{O}(\sqrt{T})$ after $T$ iterations and $\psi^T$ is $\epsilon$-EFM, then*

$$\text{NashGap}(\overline{\boldsymbol{\sigma}}^T) \leq \mathcal{O}(1/\sqrt{T}) + 2M\sqrt{2\epsilon}, \tag{2}$$

*where $M = \max_{i \in \mathcal{N}} \max_{z \in \mathcal{Z}} |u_i(z)|$.*

For a given horizon $T$, we define the meta-loss of NPCFR to be the average mutual information of the average terminal reach of the strategies selected up to $T$ on games $g \sim G$

$$\mathcal{L}(\theta) = \mathop{\mathbb{E}}_{g \in G}\left[\frac{1}{T} \sum_{t=1}^T \mathbb{I}(\psi_\theta^t)\right]. \tag{3}$$

Note minimizing this loss is different from directly minimizing the extensive form marginalizability after $T$ steps. We do this to encourage the iterates to be marginalizable as well. This is analogous to minimizing $\sum_{t=1}^{T} f(x^t)$ rather than $f(x^T)$ as in (Andrychowicz et al., 2016), where the authors meta-learned a function optimizer. The computational graph of NPCFR is shown in Figure 1. The gradient of (3) originates in the cumulative mutual information and propagates through the game tree, the regrets $r^t$, $R^t$ and the hidden states $h^t$. The gradients accumulate in the predictor $\pi(\cdot|\theta)$, which is used by the algorithms $m_\theta$ at every information state $s \in \mathcal{S}_i$ and every step $t$, see Algorithm 1.

# 4. Experiments

We conduct our experiments in general-sum games where regret minimizers are not guaranteed to converge to a Nash equilibrium. Starting in the normal-form setting, we present a distribution of games for which standard regret minimization algorithms converge to a strictly correlated CCE. We then apply our meta-learning framework to the extensive-form settings, showing we can obtain much better approximate Nash equilibria than prior algorithms. Finally, we illustrate that the meta-learned algorithms may lose their empirical performance when used out-of-distribution.

We minimize (3) for $T = 32$ iterations over 256 epochs using the Adam optimizer. The neural network uses two LSTM layers followed by a fully-connected layer. We performed a small grid search over relevant hyperparameters, see Appendix B. The meta-learning can be completed in about ten minutes for the normal-form experiments, and ten hours extensive-form games on a single CPU. See Table 4 for the memory requirements of all algorithms used.

We compare the meta-learned algorithms to a selection of current and former state-of-the-art regret minimization algorithms. Each algorithm is used to minimize counterfactual regret at each infostate of the game tree (Zinkevich et al., 2007). Specifically, we use regret matching (CFR) (Hart & Mas-Colell, 2000), predictive regret matching (PCFR) (Farina et al., 2021), smooth predictive regret matching (SPCFR) (Farina et al., 2023), discounted and linear regret minimization (DCFR, LCFR) (Brown & Sandholm, 2019b), and Hedge (Lattimore & Szepesvári, 2020). Whenever applicable, we also investigate the 'plus' version of each algorithm (Tammelin et al., 2015).

## 4.1. Normal-Form Games

The Shapley game

$$
u_1(\boldsymbol{\sigma}) = \boldsymbol{\sigma}_1^\top \cdot \begin{pmatrix} 1 & 0 & 0 \\ 0 & 1 & 0 \\ 0 & 0 & 1 \end{pmatrix} \cdot \boldsymbol{\sigma}_2,
$$

$$
u_2(\boldsymbol{\sigma}) = \boldsymbol{\sigma}_1^\top \cdot \begin{pmatrix} 0 & 1 & 0 \\ 0 & 0 & 1 \\ 1 & 0 & 0 \end{pmatrix} \cdot \boldsymbol{\sigma}_2, \tag{4}
$$

was used as a simple example where the best-response dynamics doesn't stabilize (Shapley, 1964). Indeed, it cycles on the elements which are non-zero for one player. The empirical average joint-strategy converges to a CCE

$$
\boldsymbol{\delta}^* = \frac{1}{6} \begin{pmatrix} 1 & 1 & 0 \\ 0 & 1 & 1 \\ 1 & 0 & 1 \end{pmatrix}. \tag{5}
$$

Clearly, $\boldsymbol{\delta}^*$ is not a Nash equilibrium, as it cannot be written as $\boldsymbol{\sigma}_1 \boldsymbol{\sigma}_2^\top$. However, thanks to the symmetry of the game, the marginals of $\boldsymbol{\delta}^*$, or the uniform strategy, turn out to be a Nash equilibrium.

In order to break the symmetry, we perturb the utility of one of the outcomes of the game. Specifically, we give payoff $\eta \in \mathbb{R}$ to both players when the first player selects the first, and the second their last action, see Appendix C.1. To preserve that $\boldsymbol{\delta}^*$ is a CCE, the perturbation $\eta$ needs to be bounded. We show in Appendix C.1 that for $\eta \leq 1/2$, $\boldsymbol{\delta}^*$ is a CCE. Furthermore, there is a unique Nash equilibrium, which is non-uniform for $\eta \neq 0$. We denote the distribution over biased Shapley games for $\eta \sim \mathcal{U}(a, b)$ as `biased_shapley`$(a, b)$.

To quantify the performance of the regret minimization algorithms, we study the chance that they find a solution with a given NashGap. We present our results in Table 1. All the prior regret minimization algorithms fail to reliably find the Nash equilibrium. The 'plus' non-meta-learned algorithms exhibit particularly poor performance in this regime, typically converging to a strictly correlated CCE. However, they don't all converge to $\boldsymbol{\delta}^*$ either, see Figure 2 for an illustration of the joint strategy profiles each algorithm converges to. In contrast, NPCFR$^{(+)}$ exhibit fast convergence and remarkable generalization. We show the convergence comparison of the regret minimization algorithms on `biased_shapley`$(0, 1/2)$ in Figure 4 in Appendix D.1. Despite being trained only for $T = 32$ steps, our meta-learned algorithms are able to minimize NashGap past $10^4$ steps.

| NashGap | CFR$^{(+)}$ | | PCFR$^{(+)}$ | | DCFR | LCFR | SPCFR$^{(+)}$ | | Hedge$^{(+)}$ | | NPCFR$^{(+)}$ | |
|---------|------|------|------|------|------|------|------|------|------|------|------|------|
| $10^{-2}$ | 0.78 | 0.09 | **1** | 0.09 | 0.09 | 0.42 | **1** | 0.09 | **1** | 0.36 | **1** | **1** |
| $10^{-3}$ | 0.09 | 0.02 | 0.91 | 0.02 | 0.02 | 0.02 | **1** | 0.02 | **1** | 0.06 | **1** | **1** |
| $10^{-5}$ | 0 | 0 | 0.02 | 0 | 0 | 0 | 0.11 | 0 | 0.25 | 0 | 0.14 | **1** |

*Table 1.* The fraction of games from `biased_shapley` each algorithm can solve to a given NashGap within $2^{14} = 16,384$ steps. For the algorithms marked $^{(+)}$, the left column show the standard version, while the right shows the 'plus'. See also Table 3 in Appendix D.1.

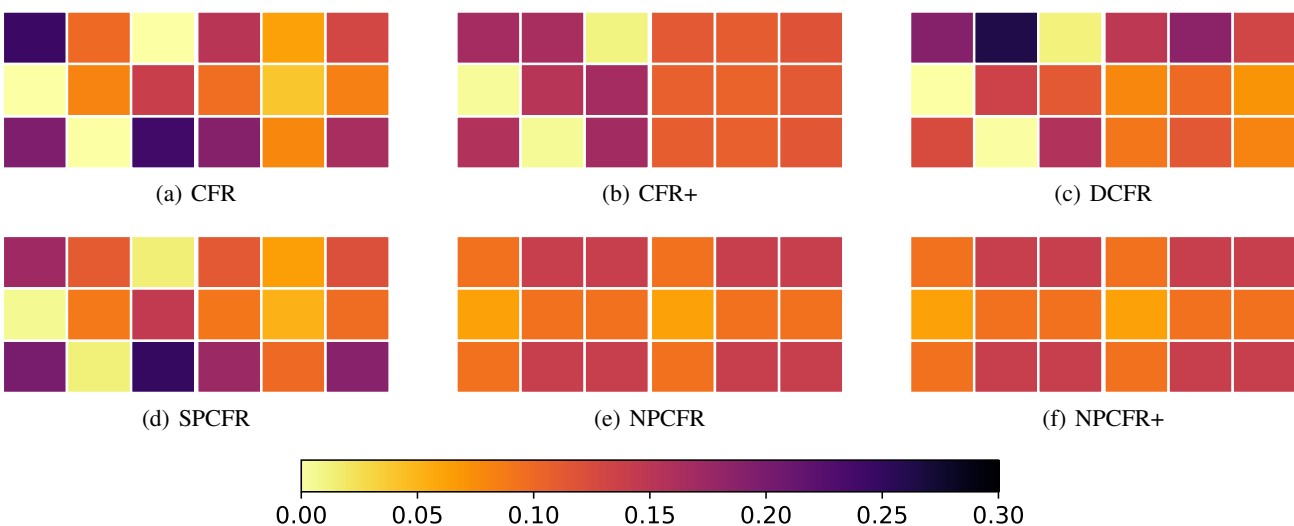

(a) CFR      (b) CFR+      (c) DCFR

(d) SPCFR      (e) NPCFR      (f) NPCFR+

0.00   0.05   0.10   0.15   0.20   0.25   0.30

*Figure 2.* The empirical average joint strategy profiles found by regret minimizers $\overline{\delta}^T$ (left) and its marginalized version (right) found on a random sample drawn from `biased_shapley`$(0, 1/2)$ after $T = 2^{14}$ steps; see Eq. (5). Darker colors indicate higher probability under $\overline{\delta}^T$, and minimal differences between left and right figures imply the joint strategy is marginalizable. The remaining algorithms are shown in Figure 5 in Appendix C.1.

## 4.2. Extensive-Form Games

To evaluate our algorithms in a sequential setting, we use the standard benchmark Leduc poker (Waugh et al., 2009), see Appendix C.2 for more details.

### 4.2.1. TWO-PLAYER LEDUC POKER

Since Leduc poker is a zero-sum game, regret minimizers are guaranteed to converge to a Nash equilibrium in the two-player version. Under standard rules, players split the pot in the case of a tie, receiving a payoff equal to their total amount bet. We break the zero-sum property by modifying tie payoffs such that players only receive a $\beta$-fraction of their bets. This change disincentives betting to increase the size of the pot, but only if the players have the same card ranks, potentially leading to correlations in players' strategies.

We define `biased_2p_leduc` as a distribution over such games, where $\beta \sim \mathcal{U}(0, 1/2)$. To quantify the performance of regret minimization algorithms, we plot the expected NashGap for each algorithm on `biased_2p_leduc` in Figure 6. While the performance averaged over the domain is similar for all algorithms, the meta-learned algorithms obtain much better approximations of Nash equilibria in each run. To show this, we investigate the chance that they find a solution with at most a given NashGap. Table 2 shows the chance for thresholds $10^{-2}, 10^{-3}$, and $10^{-5}$. With some exceptions, non-meta-learned algorithms generally fail to find a solution with a NashGap of $10^{-2}$. The 'plus' variants perform better empirically but still struggle to obtain solutions close to a Nash equilibrium as reliably as NPCFR$^{(+)}$. NPCFR$^{+}$ performs the best overall.

### 4.2.2. THREE-PLAYER LEDUC POKER

Generally, meta-learning is applied over a distribution of problem instances. However, in our setting, it is appealing even to apply it to a single instance of a game. This is because regret minimization algorithms are not guaranteed to converge to a Nash equilibrium in general-sum games. However, our meta-learning framework allows us to obtain better approximations of Nash equilibrium.

| NashGap | CFR$^{(+)}$ | | PCFR$^{(+)}$ | | DCFR | LCFR | SPCFR$^{(+)}$ | | Hedge$^{(+)}$ | | NPCFR$^{(+)}$ | |
|---|---|---|---|---|---|---|---|---|---|---|---|---|
| $10^{-2}$ | 0 | **1** | 0.03 | **1** | 0.13 | 0 | 0.54 | **1** | 0 | 0.29 | 0.84 | **1** |
| $10^{-3}$ | 0 | 0 | 0 | 0.87 | 0 | 0 | 0 | 0.72 | 0 | 0 | 0.73 | **0.98** |
| $10^{-5}$ | 0 | 0 | 0 | 0.16 | 0 | 0 | 0 | 0.11 | 0 | 0 | 0.73 | **0.96** |

*Table 2.* The fraction of games from `biased_2p_leduc` each algorithm can solve to a given NashGap within $2^{18} = 262,144$ steps. For the algorithms marked $^{(+)}$, the left column show the standard version, while the right shows the 'plus'. See also Table 5 in Appendix D.2.

We demonstrate this approach on the three-player version of Leduc poker; see Appendix C.2. We refer to the game as `three_player_leduc`. There have been conflicting reports in the literature as to the ability of regret minimization algorithms to converge to a Nash equilibrium in this game (Risk & Szafron, 2010; MacQueen & Wright, 2024). We found the performance of non-meta-learned algorithms varied significantly, with those using alternating updates giving approx. $4 - 6$-times better results. The best approximation of a Nash equilibrium we found among non-meta-learned algorithms using alternating updates[3] was NashGap $= 0.004$, produced by CFR$^+$. Without alternating updates, we found NashGap $= 0.027$, produced by CFR. Our meta-learned algorithms have been able to find a strategy with NashGap $= 0.012$ for NPCFR, and NashGap $= 0.001$ for NPCFR$^+$; see Table 6 and Figure 7 in Appendix D.3 for details. To the best of our knowledge, this is the closest approximation of Nash equilibrium of `three_player_leduc`.

To the best of our knowledge, the only theoretically sound way to find a Nash equilibrium in this game is to use support-enumeration-based algorithms such as the Lemke-Howson (Lemke & Howson, 1964). First, we would need to transform it into a two-player general-sum game. This can be done by having one of the players always best-respond, and treating them as a part of chance.[4] However, all of these algorithms work with the game in normal-form. For `three_player_leduc`, the number of pure strategies per player is $\approx 10^{472}$, making these approaches unusable in practice.

### 4.3. Out-of-Distribution Convergence

To illustrate that the meta-learned algorithms are tailored to a specific domain, we evaluate them out-of-distribution. Specifically, we run NPCFR$^{(+)}$, which were trained on `biased_shapley(0, 1/2)`, on

biased_shapley$(-1, 0)$. When evaluated out-of-distribution, the meta-learned algorithms lose the ability to converge to a Nash equilibrium. See Figure 8 in Appendix D.4 for more details.

## 5. Conclusion

We present a novel framework for approximating Nash equilibria in general-sum games. We apply regret minimization, which is a family of efficient algorithms, guaranteed to converge to a coarse-correlated equilibrium (CCE). This weaker solution concept allows player to correlate their strategies. We use meta-learning to search a class of predictive regret minimization algorithms, minimizing the correlations in the CCE found by the algorithm. The resulting algorithm is still guaranteed to converge to a CCE, and is meta-learned to empirically find close approximations of Nash equilibria. Experiments in general-sum games, including large imperfect-information games, reveal our algorithms can considerably outperform other regret minimization algorithms.

**Future Work.** Our meta-learning framework might be useful for finding CCEs with desired properties. For example, one can search for welfare maximizing equilibria by setting the meta-loss to the negative total utility of all players. We also see other domains, such as auctions, as a promising field where our approach can be used. One limitation of our approach is that it can be quite memory demanding, especially for larger horizons. Training on abstractions of the games is promising.

## Impact Statement

This paper presents work whose goal is to advance the field of Machine Learning. There are many potential societal consequences of our work, none which we feel must be specifically highlighted here.

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
