# OpenReview forum: "Approximating Nash Equilibria in General-Sum Games via Meta-Learning"
_ICML.cc/2025/Conference — Submitted to ICML 2025_

### Official Review · Reviewer_PwHH · 2025-02-16

**Overall Recommendation:** 2

**Summary:**

Finding exact Nash equilibria (NE) in general-sum games is known to be PPAD-complete. In contrast, regret minimization is a well-established method for learning equilibrium strategies, but it only guarantees convergence to coarse correlated equilibria (CCE; a weaker equilibrium notation than Nash equilibrium) in general-sum games. This submission introduces a meta-learning framework that adapts regret minimization to minimize the correlation in players’ strategies, thereby pushing the CCEs closer to NEs.

The authors build off of the “learning not to regret” framework of Sychrovksy et al. 2024, which uses meta-learning to improve convergence speed in games. Unlike classical regret minimization algorithms that treat strategy updates as independent, the authors proposed methods meta-learn a predictive model that adjusts the regret updates to minimize strategic correlation. They provide a bound on the distance to NE when using their approach, and evaluate their methods on both normal-form games and poker subgames.

**Claims And Evidence:**

Yes.

**Essential References Not Discussed:**

This work is missing a discussion on the broader literature on meta-learning, as well as the reference "Meta-Learning in Games" by Harris el al., published in ICLR 2023.

**Experimental Designs Or Analyses:**

Yes.

**Methods And Evaluation Criteria:**

Yes.

**Other Comments Or Suggestions:**

n/a

**Other Strengths And Weaknesses:**

Strengths:
This submission extends the application of meta-learning techniques in game theory by training a regret minimization algorithm to approximate NE in general-sum games. By designing a meta-loss function that penalizes correlation, the authors introduce a principled way to reduce deviations from NE. Unlike standard regret minimization, which only guarantees convergence to CCE, the authors provide a bound on the distance to a NE in terms of their meta-objective. They introduce the concept of extensive-form marginalizability, and prove that reducing mutual information between players’ strategies leads to better NE approximations.

Weaknesses:
While the paper provides a bound on the distance to NE, their approach does not guarantee convergence to a NE. While this is itself not a weakness, it would have been nice to see more discussion surrounding this result (e.g. where it produces a good approximation, when it fails), as it is the main theoretical contribution of this work and I was a bit confused about what the takeaway should be. Empirically, while the authors’ approach improves performance in some settings, it performs inconsistently in others. For example, their methods are out-performed by Hedge (a very simple algorithm) in terms of convergence to small NashGap in the biased_shapley game. Finally, the use of neural networks increases the computational costs per iteration.

**Questions For Authors:**

n/a

**Relation To Broader Scientific Literature:**

This work builds on work in meta-learning in games. Specifically, they use meta-learning techniques to converge to CCEs that are closer to NEs.

**Theoretical Claims:**

Yes.

---

> ### Author Rebuttal · Authors · 2025-03-31
>
> We would like to sincerely thank the reviewers for their time spent to help improve our work.
> We appreciate all the comments and will integrate them into a revised version of the paper.
> Let us address the questions and comments raised.
>
> The main theoretical conclusion of our paper is that if one can efficiently minimize the mutual information loss, then the resulting strategy of NPCFR is close to a NE.
> However, assuming P$\neq$PPAD, there are games for which one cannot efficiently minimize the loss.
> As we have shown in our experiments, our approach works well in practice on standard benchmarks.
> We are not sure where our approach would be guaranteed to fail, as that would require finding the games for which one cannot find NE in polynomial-time, proving P$\neq$PPAD.
>
> It is true that one version of Hedge does outperform one variation of our NPCFR in biased shapley.
> We wanted to show that while some of the classic algorithms can get closer to the NE, none of them converges to it.
> On the other hand, our NPCFR+ reliably converges to NE withing machine precision, see Figure 4 in the Appendix.
>
> Our algorithm does indeed have higher computation cost given the use of neural networks.
> However, we want to stress that there are no other algorithms which can be even applied to the games we consider in our experiments and would be guaranteed to converge to NE.
>
> Regarding the reference you mentioned, the paper considers meta-learning in a repeated setting, trying to leverage the past solved games to improve convergence rate on the next one.
> In our setting, we do meta-learning offline, and fix the regret minimizer at test-time.
> Furthermore, their results don't apply to the problem of approximating Nash equilibria, which is our main focus.

---

> > ### Comment · Reviewer_PwHH · 2025-04-01
> >
> > Some of the results in "Meta-Learning in Games" do apply to the problem of approximating Nash equilibria (albeit only in zero-sum games). See, e.g., Theorem 3.2.
> >
> > Do you know of any sub-classes of games where one can efficiently minimize the mutual information loss?

---

> > > ### Author Response · Authors · 2025-04-04
> > >
> > > Yes, that is true. We will make sure to add it to the related work.
> > >
> > > Any class of games where one can find a Nash equilibrium in polynomial time admits efficient (in the polynomial sense) minimization of the mutual information loss. This would include for example zero-sum and potential games. However, being able to efficiently minimize the loss implies being able to efficiently approximate Nash, which is known to be computationally hard. We have not extended the class polynomial-time solvable games in this paper in this way.

---

### Official Review · Reviewer_Gvh1 · 2025-02-19

**Overall Recommendation:** 3

**Summary:**

This paper falls into the approximation of NEs in general-sum, normal form, or especially extensive-form games.
While regret-minimization algorithms only guarantee convergence to a CCE, this paper proposes NPCFR, which uses neural networks to parameterize such an algorithm.
The parameterization is hard-coded to be bounded, thus the parameterized algorithm preserves regret-minimization guarantee and converges to a CCE.
Besides, the network parameters are optimized to minimize the mutual information of the approximated CCE (up to T=32 algorithm iteration), for which zero mutual information indicates that the CCE is also an NE.
The paper finally shows the effectiveness of NPCFR on two experiments, one is a standard normal-form game and the other is a poker game.

**Claims And Evidence:**

There are some problematic claims in this paper.

* In line 134-135 RHS, the author claims that "A joint strategy profile δ is a CCE if and only if its CCE Gap is zero." Actually, CCE Gap can be negative in the following example: two players with two actions each, where the payoff matrix is:
$$
\begin{bmatrix}
1, 0
\end{bmatrix}
$$
$$
\begin{bmatrix}
0, 1
\end{bmatrix}
$$
 for each player. The correlated strategy that puts 0.5 probability on two 1-element constitutes a CCE, but the CCE Gap is -0.5.

* In line 414-417 LHS, the author claims "To the best of our knowledge, the only theoretically sound way to find a Nash equilibrium in extensive-form games is to use support-enumeration-based algorithms." Actually, to solve an extensive-form game, one does not need to transform it into a normal-form game with exponentially many pure strategies. We suggest the authors check Section 3.7-3.12 of (Nisan et al., 2007), which also appears in the paper's reference.

**Essential References Not Discussed:**

N/A.

**Experimental Designs Or Analyses:**

I've checked all experiments. Following are some issues.

* In Section 4.2.1, the author choose the two-player game with parameter $β ∼ U(0, 1/2)$, but it seems that $\beta \sim U(0,1)$ is a more natural choice since $\beta=1$ represents the game is zero-sum. Why do the authors exclude the case $\beta \in [1/2, 1]$?
* The paper does not compare non-regret-minimization algorithms, which might be more efficient in two experiments in this paper. The paper also does not compare NPCFR with NPRM proposed in (Sychrovsk´y et al., 2024).

**Methods And Evaluation Criteria:**

Yes.

**Other Comments Or Suggestions:**

Although this paper shares some novel ideas, the overall contribution seems to be incremental for prior works (Sychrovsk´y et al., 2024). I would like to reevaluate this paper if the authors could address my concerns about the similarity between NPCFR and (Sychrovsk´y et al., 2024).

There are also some typos in notations and some notations are not stated in mathematical rigors. These issues do not affect my understanding, but I suggest the authors carefully polish the notations in the next version. Many issues consist of:

* In line 128, the notation $w^0$ seems to be the same concept with $w^o$ in Definition 2.1.
* In line 139-145, the notation of observation function $\mathcal{O}_i$ and $\mathcal{S}_i$ lacks mathematical rigors. What is the domain and image of $\mathcal{O}_i$? What's the expression of $\mathcal{S}_i$? What's the relation between $\mathcal{O}_i$ and $\mathcal{S}_i$?
* In line 146-147, the concept $\mathcal{A}_i(s_i)$ has never appeared before.
* In line 151-152, what is the form of $\rho$? Is $\rho_i: \mathcal{S}_i \to \Delta \mathcal{A}_i(s_i)$ or $\rho_i: \mathcal{S}_i \to \mathcal{A}_i(s_i)$?
* In line 163-164, the notation $\mathbb{E}_{z\sim \delta}$ means the reach probability of $z$ with joint strategy $\delta$. This notation conflicts with the common usage of $z\sim \delta$.
* In RHS of line 113-114, the notation $\delta(\rho_i, \rho_{-i})$ is again undefined.
* In line 172-173, the expression $\max _{a\in \mathcal{A} _i} \textbf{R}^T _i$ should be $\max _{a\in \mathcal{A} _i} R^T _i(a)$ or $\max _{a\in \mathcal{A} _i} R^T _{i,a}$, I guess.
* In Figure 2, the author claims the game is randomly drawn from biased_shapley(0,1/2). From the visualization provided, it seems that the parameter is close to $1/2$. I suggest the author clarify which parameter is used in this figure.

**Other Strengths And Weaknesses:**

Strengths:
S1: The paper is well-written and easy to follow.
S2: The meta-learning procedure is novel and interesting.
S3: Theorem 1 serves as a theoretical foundation for the advantages of minimizing mutual information.

Weaknesses:
W1: The methodology proposed in this paper seems to be restricted by the size of the extensive-form game, and the nodes $S_i$ in the infostate tree might be exponential on the number of players and time periods (in extensive-form games). These issues may hinder NPCFR from scaling to larger extensive-form games.
W2: NPCFR seems to be similar to NPRM proposed in (Sychrovsk´y et al., 2024), with the only difference on the objective function that is used to train network parameters.

**Questions For Authors:**

Q1: How are NashGap and $\mathbb{I}(\psi)$ computed or approximated in extensive-form games? It seems that in the original definition in RHS of line 117-118, and the RHS of line 255, these definitions are hard to compute since the corresponding extensive-form games have exponential size w.r.t. game description (say, player number $n$ and time period in extensive-form games).

**Relation To Broader Scientific Literature:**

Existing works show that regret-minimization algorithms guarantee to converge to CCEs but not NEs.
The related paper (Sychrovsk´y et al., 2024) utilizes neural networks to parameterize a class of regret-minimization algorithms, in which network parameters are trained to predict next-iteration regret.
This paper's contribution lies in that the network parameters are trained w.r.t. mutual information minimization of the approximated CCE (through $T=32$ iterations of the network-parameterized regret-minimization algorithm).

**Theoretical Claims:**

I take a casual glance at the proof of Theorem 1, and it seems correct as far as I can tell.

---

> ### Author Rebuttal · Authors · 2025-03-31
>
> We would like to sincerely thank the reviewers for their time spent to help improve our work.
> We appreciate all the comments and will integrate them into a revised version of the paper.
> Let us address the questions and comments raised.
>
> Thank you for pointing out the negative CCE Gap example, we agree and change the condition to CCE Gap being non-positive.
>
> Regarding the sequence for representation, it would indeed be very interesting to modify algorithms such as the Lemke-Howson to work with it.
> However, since we are not aware of any such algorithm in the literature, we opted not to pursue this and leave it for future work.
>
> We chose to avoid values close to $\beta=1$ because we wanted to study ``strictly general-sum'' games, where the regret minimizers don't have the convergence guarantees to NE.
> However, including the other games empirically leads to similar results.
>
> In the normal-form setting, algorithms such as Lemke-Howson would indeed solve all instances fast.
> However, biased shapley is not the intended use case of our algorithms, but rather just a toy example showing that all standard regret minimizers fail to find NE.
> In the extensive-form case, unless we would have a way to work in the sequence-form representation, the classical algorithms cannot even be initialized.
>
> Our NPCFR is a special version of NPRM introduced in Sychrovsky 2024, which is tailored to find NE in general-sum games.
> Using NPRM, which is only trained to converge fast to a CCE, would likely cause it to fail even in biased shapley.
> It is this meta-loss which enables us to recover convergence to NE in biased shapley.
> More importantly, it allows us find the to our best knowledge closest approximation of NE in three-player Leduc poker, a standard benchmark used in the field of general-sum games.
> Together with our theoretical analysis, we believe that this is a major advancement in the field of approximating Nash equilibria.
>
> Thank you for pointing out the parts of the paper which need to be made more precise.
> Due to the tight space constraints, we tried to follow standard notation defined, for example, in (Kovarik et al., 2022) and clarify whenever possible.
> We plan to include a section in the Appendix to elaborate on the notation.
> To address your questions
>
> 1) Yes, that is a typo, thank you.
> 2) $\mathcal{O}_i$ maps the sequence of actions that lead to a world state to observations of player $i$. World states, for which the observation function gives the same output, form the infostate $s_i\in\mathcal{S}_i$.
> 3) We abuse the notation a little bit here to define the set of actions available to player $i$ in $s_i$.
> 4) $\rho_i$ is a deterministic strategy. The utility is defined over distributions on such deterministic strategies.
> 5) We identify the terminal state with a pure strategy profile. In this sense, the joint strategy profile is a distribution over terminal states.
> 6) Yes, the cumulative regret vector is indexed by actions, we will make it more clear.
> 7) Yes, the sample corresponds to $\eta \approx 0.42$, we will include it in the Figure.
>
> To compute the mutual information loss in the extensive-form, we compute the reach probability of each terminal state given the strategy in the tree.
> We than factorize it per-player, and compute the outer product.
> Finally, we compute the KL-divergence between the two distributions.
> This is a natural extension of the normal-form case, where the pairs of actions correspond to the terminal states.

---

### Official Review · Reviewer_jLNx · 2025-03-03

**Overall Recommendation:** 2

**Summary:**

This work provides a meta-learning strategy to guarantee convergence to a CCE (with low correlation) of no-regret learners for a repeatedly played n-player general-sum game.

**Claims And Evidence:**

Proofs for theoretical results are provided. However, I am not able to verify these proofs. (see questions to the authors)

**Essential References Not Discussed:**

References discussed sufficiently.

**Ethical Review Concerns:**

none.

**Experimental Designs Or Analyses:**

This paper is primarily theoretical work. The illustrating experiments seem reasonable to me.

**Methods And Evaluation Criteria:**

This is primarily theoretical work. The illustrating experiments do make sense.

**Other Comments Or Suggestions:**

none.

**Other Strengths And Weaknesses:**

I am having problems understanding the claims and verifying the proofs. I might miss something, but overall I got the impression that this paper (although it may contain a nice idea) is not yet polished enough to be published.

Please, see the questions to the authors for some details on points which were unclear to me.

**Questions For Authors:**

There are several points of the paper I do not understand:

(1) What assumptions are made on the utilities $u_i$?

(2) Does the argmax for the definition of the best-responds exist? Related to this: what are the assumptions for $S_i, A_i$ and $u_i$?

(3) On page 4, line 198, it is stated that '[...] In two-player zero-sum games, the empirical average strategy $\bar \sigma$ is guaranteed to converge to a Nash equilibrium.' The paper references Zinkevich et al '07. From which statement/result in Zinkevich et al '07 does this claim follow?

(4) Are the action spaces $A_i$ assumed to be finite, convex, closed or bounded?

(5) How is $S_i$ defined exactly?

(6) From the definition, I understand that $x_i^t$ is a vector (finite-dimensional?). How is $x_i^t(\sigma^t)$ defined? And what is $\langle \sigma_i^t(s), x_i^t(\sigma^t) \rangle 1$?

**Relation To Broader Scientific Literature:**

This work falls into the general line of research on approximating CCEs.

**Theoretical Claims:**

The theoretical claims of this paper are hard to follow and verify due to missing formal definitions and unclear assumptions.  See questions to the authors.

---

> ### Author Rebuttal · Authors · 2025-03-31
>
> We would like to sincerely thank the reviewers for their time spent to help improve our work.
> We appreciate all the comments and will integrate them into a revised version of the paper.
> Let us address the questions and comments raised.
>
> 1) We make no assumptions about the utility function. The regret minimization algorithms are guaranteed to find CCEs in any general-sum game.
> 2) Yes, the domain of the strategy $\sigma_i$ is bounded and closed, and one can thus use the Extreme value theorem. The joint action space $\mathcal{A}$ is the set of all sequences of actions the players can perform in the game. For example, in poker check-bet-fold is an element of $\mathcal{A}$. The elements of $\mathcal{S}_i$ are called infostates, and are used to model the information the player has at that stage of the game, and which she can condition her strategy on.
> 3) Their Theorem 2, in combination with the vanishing average regret of a regret minimization algorithm, proves the claim.
> 4) The joint action set is discrete and finite. One can build a continuous set of strategies, which are distributions over actions available to players, which is bounded, convex and closed similar to other distributions.
> 5) The action-observation set $\mathcal{S}_i$ consists of histories, which are sequences of action-observation pairs from the beginning of the game. In case of imperfect-information games, some action-observation pairs are modified by the observation function to mask private information of the other players.
> 6) Yes, $x_i$ is a vector of rewards per-action for player $i$. $x_i(\sigma)$ are rewards of $i$ when players use strategy $\sigma = (\sigma_1, \dots \sigma_n)$. $\langle\sigma_i,x_i\rangle$ is the inner product, and $\mathbf{1} = (1, \dots 1)$ is a vector of $|x_i|$ ones.

---

### Official Review · Reviewer_EMmS · 2025-03-14

**Overall Recommendation:** 3

**Summary:**

The paper proposes a new method for finding approximate Nash equilibria of n-player extensive-form games. The method uses regret learners. Regret learners can be proved to converge to coarse-correlated equilibria (CCE). To make the CCE closer to Nash, the paper proposes that we train a regret minimizer. The paper gives a theoretical result bounding the Nash gap of the resulting optimizer, and demonstrates the effectiveness of the method relative to normal regret minimizers.

**Claims And Evidence:**

Yes.

**Essential References Not Discussed:**

I can’t think of any, though I do wonder whether there really isn’t more existing work on Nash equilibria in extensive-form games.

**Experimental Designs Or Analyses:**

I only checked what’s in the main text of the paper.

**Methods And Evaluation Criteria:**

Generally yes.

That said, one question I have is: Why is there no comparison to, say, Lemke-Howson (or other Nash-equilibrium-finding algorithms) for the normal-form case?

**Other Comments Or Suggestions:**

>The meta-learning can be completed in about ten minutes for the normal-form experiments, and ten hours extensive-form games on a single CPU.

Missing “for the” between “hours” and “extensive-form”.

**Other Strengths And Weaknesses:**

Generally the contributions seem strong, despite the fact that the meta learning idea is mostly from prior work.

Unfortunately, I’m not in a great position to judge this paper.

**Questions For Authors:**

See “Methods And Evaluation Criteria”.

**Relation To Broader Scientific Literature:**

On the one hand, the paper relates to the computational problem of finding or approximating Nash equilibria of general-sum extensive-form games. It also builds on regret learners specifically, which have been found to often converge to Nash equilibria. It also builds on the idea of (meta-)training of Sychrovský et al. 2024.

**Theoretical Claims:**

I didn’t check.

---

> ### Author Rebuttal · Authors · 2025-03-31
>
> We would like to sincerely thank the reviewers for their time spent to help improve our work.
> We appreciate all the comments and will integrate them into a revised version of the paper.
> Let us address the questions and comments raised.
>
> The reason why we don't compare with algorithms such as Lemke-Howson is that, at least for the extensive-form games we study, they cannot even be initialized.
> This is because they require the game to be in normal-form, which is exponentially larger than the extensive-form.
> In the case of biased shapley, it is hard to compare Lemke-Howson with the regret minimization algorithms, because the notion of ``iteration'' is very different between the two.
> Other natural metrics, such as running time, are very implementation and hardware specific.

---

> > ### Comment · Reviewer_EMmS · 2025-04-04
> >
> > (Sorry the relatively late reply!)
> >
> > (Note that my question about comparison to Lemke-Howson was indeed specifically about comparing the algorithms in the _normal-form_ case.)
> >
> > I agree that the difference in what happens during each iterations makes comparisons between Lemke-Howson and regret-based methods meaningless. I was indeed thinking more of something like running time.
> >
> > On whether a running time comparison, I'm sympathetic to the authors' response, although perhaps not entirely convinced.
> >
> > One the hand, I do think that fundamentally different algorithms can be reasonably compared. For instance, we can compare different sorting algorithms empirically. One consideration in favor of the feasibility of such comparisons is that if one algorithm is fundamentally better, then it'll be better even if hardware somewhat favors the other algorithm or if the implementation is slightly inefficient.
> >
> > On the other hand, I do agree that there are lots of issues related to choosing a specific implementation. For instance, I assume that reference implementations for Lemke-Howson would be highly optimized and thus difficult to compete against and in my opinion, the authors souldn't have to solve all the practical optimization issues to publish this paper. Writing your own Lemke-Howson implementation to compare against has its own issues. Still, maybe there is some "basic" reference implementation of Lemke-Howson? (I suspect the nashpy implementation isn't highly optimized, for instance.)
> >
> > Anyway, I should also say that I'm not the right person to judge the practicality of this sort of comparison.

---

> > > ### Author Response · Authors · 2025-04-05
> > >
> > > Thank you for you response.
> > >
> > > We see the main contribution of our paper in the extensive=form applications, where it pushes the boundary of where the Nash equilibrium can be approximated. While we treat the normal-form case as a toy-example, we are confident that our approach would scale better to larger games. While this is likely a fruitful direction of future research, we felt it was out of scope of this paper.
> > >
> > > For the normal-form games we study in the paper, we are confident that e.g. the nashpy implementation would be faster.

---

### Decision · Program_Chairs · 2025-05-01

**Decision:**

Reject

**Comment:**

Reviewers generally liked the approach while raising concerns on significance, similarity to [Sychrovsky et al., 2024], experiments, and clarity. Unfortunately no reviewer became excited enough to champion the paper. We hope the authors find the reviews helpful. Thanks for submitting to ICML!